# Tensor Monte Carlo: Particle Methods for the GPU era

**Laurence Aitchison**[*]
University of Bristol
Bristol, UK
laurence.aitchison@gmail.com

## Abstract

Multi-sample, importance-weighted variational autoencoders (IWAE) give tighter bounds and more accurate uncertainty estimates than variational autoencoders (VAEs) trained with a standard single-sample objective. However, IWAEs scale poorly: as the latent dimensionality grows, they require exponentially many samples to retain the benefits of importance weighting. While sequential Monte-Carlo (SMC) can address this problem, it is prohibitively slow because the resampling step imposes sequential structure which cannot be parallelised, and moreover, resampling is non-differentiable which is problematic when learning approximate posteriors. To address these issues, we developed tensor Monte-Carlo (TMC) which gives exponentially many importance samples by separately drawing $K$ samples for each of the $n$ latent variables, then averaging over all $K^n$ possible combinations. While the sum over exponentially many terms might seem to be intractable, in many cases it can be computed efficiently as a series of tensor inner-products. We show that TMC is superior to IWAE on a generative model with multiple stochastic layers trained on the MNIST handwritten digit database, and we show that TMC can be combined with standard variance reduction techniques.

Variational autoencoders (VAEs) (Kingma & Welling, 2013; Rezende et al., 2014; Eslami et al., 2018) have had dramatic success in exploiting modern deep learning methods to do probabilistic inference in previously intractable high-dimensional spaces. However, standard VAEs using a single-sample objective give loose variational bounds and poor approximations to the posterior (Turner & Sahani, 2011; Burda et al., 2015). Modern variational autoencoders instead use a multi-sample objective to improve the tightness of the variational bound and the quality of the approximate posterior (Burda et al., 2015). These methods implicitly improve the approximate posterior by drawing multiple samples from a proposal, and resampling to discard samples that do not fit the data (Cremer et al., 2017).

While multi-sample importance-weighted methods are often extremely effective, they scale poorly with problem size. In particular, recent results (Chatterjee & Diaconis, 2015) have shown that the number of importance samples required to closely approximate any target expectation scales as $\exp(D_{\mathrm{KL}}(\mathrm{P}||\mathrm{Q}))$, where Q is the proposal distribution (and this was prefigured by earlier work in the particle filter context (Snyder et al., 2008; Bengtsson et al., 2008)). Critically, the KL-divergence scales roughly linearly in problem size (and exactly linearly if we consider $n$ independent sub-problems being combined), and thus we expect the required number of importance samples to be exponential in the problem size. As such, multi-sample methods are typically only used to infer the latent variables in smaller models with only a single (albeit vector-valued) latent variable (e.g. Burda et al., 2015).

---

[*]Work done while at Janelia Research Campus.

One approach to resolving these issues is sequential Monte-Carlo (SMC) (Maddison et al., 2017; Naesseth et al., 2017; Le et al., 2017), which circumvents the need for exponentially many samples using resampling. However, SMC has two issues. First, the SMC resampling steps force an inherently sequential structure on the computation, which can prohibit effective parallelisation on modern GPU hardware. While this is acceptable in a model (such as a state-space model) that already has sequential structure, SMC has been applied in many other settings where there is considerably more scope for parallelisation such as mixture models (Fearnhead, 2004) or even probabilistic programs (Wood et al., 2014). Second, modern variational inference uses the reparameterisation trick to obtain low-variance estimates of the gradient of the objective with respect to the proposal parameters (Kingma & Welling, 2013; Rezende et al., 2014). However, the reparameterisation trick requires us to differentiate samples from the proposal with respect to parameters of the proposal, and this is not possible in SMC due to the inherently non-differentiable resampling step (and this is true even in variants such as particle filtering with backward sampling Doucet & Johansen (2009)). As such, while it may be possible in some circumstances to obtain reasonable results using a biased gradient (Maddison et al., 2017; Naesseth et al., 2017; Le et al., 2017), those results are empirical and hence give no guarantees.

To resolve these issues, we introduce tensor Monte Carlo (TMC). While standard multi-sample objectives draw $K$ samples from a proposal over all latent variables jointly, TMC draws $K$ samples for each of the $n$ latent variables separately, then forms a lower-bound by averaging over all $K^n$ possible combinations of samples for each latent variable. To perform these averages over an exponential number of terms efficiently, we exploit conditional independence structure in a manner that is very similar to early work on graphical models (Pearl, 1986; Lauritzen & Spiegelhalter, 1988). In particular, we note that for TMC, as well as for classical graphical models, these sums can be written in an extremely simple and general form: as a series of tensor inner products. As such, TMC is most closely related to an MCMC method known as the "embedded HMM" (Neal et al., 2004) which performs MCMC sampling by drawing $K$ samples for each latent, and sampling from the resulting $K^n$ state space.

Finally, as TMC is, in essence, IWAE with exponentially many importance samples, it can be combined with previously suggested variance reduction techniques, including (but not limited to) sticking the landing (STL) (Roeder et al., 2017), doubly reparameterised gradient estimates (DReGs) (Tucker et al., 2018), and reweighted wake-sleep (RWS) (Bornschein & Bengio, 2014; Le et al., 2018).

# 1   Background

Classical variational inference consists of optimizing a lower bound, $\mathcal{L}_{\text{VAE}}$, on the log-marginal likelihood, $\log \text{P}(x)$,

$$\log \text{P}(x) \geq \mathcal{L}_{\text{VAE}} = \log \text{P}(x) - D_{\text{KL}}\left(\text{Q}(z)\,||\,\text{P}(z|x)\right), \tag{1}$$

where $x$ is the data, $z$ is the latent variable, P is the generative model, and Q is known as either the approximate posterior, the recognition model or the proposal distribution. As the Kullback-Leibler (KL) divergence is always positive, we can see that the objective is indeed a lower bound, and if the approximate posterior, $\text{Q}(z)$, is sufficiently flexible, then as we optimize $\mathcal{L}_{\text{VAE}}$ with respect to the parameters of the approximate posterior, the approximate posterior will come to equal the true posterior, at which point the KL divergence is zero, so optimizing $\mathcal{L}_{\text{VAE}}$ reduces to optimizing $\log \text{P}(x)$. However, in the typical case where $\text{Q}(z)$ is a more restrictive family of distributions, we obtain biased estimates of the generative parameters and approximate posteriors that underestimate uncertainty (Minka et al., 2005; Turner & Sahani, 2011).

This issue motivated the development of more general lower-bound objectives, and to understand how these bounds were developed, we need to consider an alternative derivation of $\mathcal{L}_{\text{VAE}}$. The general approach is to take an unbiased stochastic estimate of the marginal likelihood, denoted $\mathcal{P}$,

$$\text{P}(x) = \text{E}_{\text{Q}}[\mathcal{P}] \tag{2}$$

and convert it into a lower bound on the log-marginal likelihood using Jensen's inequality,

$$\log \text{P}(x) \geq \mathcal{L} = \text{E}_{\text{Q}}[\log \mathcal{P}]. \tag{3}$$

We can obtain most methods of interest, including single-sample VAE's, multi-sample IWAE, and TMC by making different choices for $\mathcal{P}$ and Q. For the single-sample variational objective we use a

proposal, $Q(z)$, defined over a single setting of the latents,

$$\mathcal{P}_{\text{VAE}} = \frac{P(x, z)}{Q(z)}, \tag{4}$$

which gives rise to the usual variational lower bound, $\mathcal{L}_{\text{VAE}}$. However, this single-sample estimate of the marginal likelihood has high variance, and hence a gives a loose lower-bound. To obtain a tigher variational bound, one approach is to find a lower-variance estimate of the marginal likelihood, and an obvious way to reduce the variance is to average multiple independent samples of the original estimator,

$$\mathcal{P}_{\text{IWAE}} = \frac{1}{K} \sum_{k=1}^{K} \frac{P(x, z^k)}{Q(z^k)}, \tag{5}$$

which indeed gives rise to a tighter, importance-weighted bound, $\mathcal{L}_{\text{IWAE}}$ (Burda et al., 2015).

## 2   Results

First, we give a proof showing that we can obtain unbiased estimates of the model-evidence by dividing the full latent space into several different latent variables, $z = (z_1, z_2, \ldots, z_n)$, drawing $K$ samples for each individual latent and averaging over all $K^n$ possible combinations of samples. We then give a method for efficiently computing the required averages over an exponential number of terms using tensor inner products. We give toy experiments, showing that the TMC bound approches the true model evidence with exponentially fewer samples than IWAE, and in far less time than SMC. Finally, we do experiments on VAE's with multiple stochastic layers trained on the MNIST handwritten digit database. We show that TMC can be used to learn recognition models, that it can be combined with variance reduction techniques such as STL (Roeder et al., 2017) and DReGs (Tucker et al., 2018), and is superior to IWAE's given the same number of particles, despite negligable additional computational costs.

### 2.1   TMC for factorised proposals

In TMC we consider models with multiple latent variables, $z = (z_1, z_2, \ldots, z_n)$, so the generative and recognition models can be written as,

$$P(x, z) = P(x, z_1, z_2, \ldots, z_n) \qquad\qquad Q(z) = Q(z_1) Q(z_2) \cdots Q(z_n) \tag{6}$$

where we use a factorised proposal to simplify the proof (see Appendix B for a proof for non-factorised proposals). For the TMC objective, each individual latent variable, $z_i$, is sampled $K_i$ times,

$$z_i^{k_i} \sim Q(z_i), \tag{7}$$

where $i \in \{1, \ldots, n\}$ indexes the latent variable and $k_i \in \{1, \ldots, K_i\}$ indexes the sample for the $i$th latent. Importantly, any combination of the $k_i$'s can be used to form an unbiased, single-sample estimate of the marginal likelihood. Thus, for any $k_1, k_2, \ldots, k_n$ we have,

$$P_\theta(x) = E_{Q(z)} \left[ \frac{P(x, z_1^{k_1}, z_2^{k_2}, \ldots, z_n^{k_n})}{Q(z_1^{k_1}) Q(z_2^{k_2}) \cdots Q(z_n^{k_n})} \right]. \tag{8}$$

The average of a set of unbiased estimators is another unbiased estimator. As such, averaging over all $K^n$ settings for the $k_i$'s (and hence over $K^n$ unbiased estimators), we obtain a lower-variance unbiased estimator,

$$\mathcal{P}_{\text{TMC}} = \frac{1}{\prod_i K_i} \sum_{k_1, k_2, \ldots, k_n} \frac{P(x, z_1^{k_1}, z_2^{k_2}, \ldots, z_n^{k_n})}{Q(z_1^{k_1}) Q(z_2^{k_2}) \cdots Q(z_n^{k_n})}, \tag{9}$$

and this forms the TMC estimate of the marginal likelihood.

## 2.2 Efficient averaging

The TMC unbiased estimator in Eq. (9) involves a sum over exponentially many terms, which may be intractable. To evaluate the TMC marginal likelihood estimate efficiently, we therefore need to exploit structure in the graphical model. For instance, for a directed graphical model, the joint-probability can be written as a product of the conditional probabilities,

$$P(x, z_1, z_2, \ldots, z_n) = P\big(x|z_{\mathrm{pa}(x)}\big) \prod_{i=1}^n P\big(z_i|z_{\mathrm{pa}(z_i)}\big) \tag{10}$$

where $\mathrm{pa}(z_i) \subset \{1, \ldots, n\}$ is the indicies of the parents of $z_i$. In the case of a directed graphical model, we can write the importance ratio as a product of factors (Kschischang et al., 2001; Frey, 2003; Bishop, 2006),

$$\prod_j f_j^{\kappa_j} = \frac{P\big(x, z_1^{k_1}, z_2^{k_2}, \ldots, z_n^{k_n}\big)}{Q\big(z_1^{k_1}\big) Q\big(z_2^{k_2}\big) \cdots Q\big(z_n^{k_n}\big)} \tag{11}$$

where, $\kappa_j$'s are tuples containing the indicies ($k_i$'s) of each factor. For the model in Eq. (10), a typical choice of factors, $f_j^{\kappa_j}$ and corresponding $\kappa_j$ would be to use the $j = 0$th factor for the data,

$$f_0^{\kappa_0} = P\Big(x|z_{\mathrm{pa}(x)}^{k_{\mathrm{pa}(x)}}\Big) \qquad\qquad \kappa_0 = k_{\mathrm{pa}(x)} \tag{12}$$

and to use the $j$th factor (with $j > 0$) for the $j$th latent variable,

$$f_j^{\kappa_j} = P\Big(z_j^{k_j}|z_{\mathrm{pa}(z_j)}^{k_{\mathrm{pa}(z_j)}}\Big) \Big/ Q\Big(z_j^{k_j}\Big) \qquad\qquad \kappa_j = \big(k_j, k_{\mathrm{pa}(z_j)}\big) \tag{13}$$

As such, substituting Eq. (11) into Eq. (9), the TMC marginal likelihood estimator can be written as summations over a series of tensors, $f_j^{\kappa_j}$,

$$\mathcal{P}_{\mathrm{TMC}} = \frac{1}{K_1 K_2 \cdots K_n} \sum_{k_1, k_2, \ldots, k_n} \prod_j f_j^{\kappa_j}. \tag{14}$$

If there are sufficiently many conditional independencies in the graphical model, we can compute the TMC marginal likelihood estimate efficiently by swapping the order of the product and summation.

## 2.3 Non-factorised proposals

For non-factorised proposals, we obtain a result similar to that for factorised proposals (Eq. 9),

$$\mathcal{P}_{\mathrm{TMC}} = \frac{1}{\prod_i K_i} \sum_{k_1, k_2, \ldots, k_n} \frac{P\big(x, z_1^{k_1}, z_2^{k_2}, \ldots, z_n^{k_n}\big)}{\prod_i Q\big(z_i^{k_i}|\mathbf{z}_{\mathrm{qa}}(z_i)\big)} \tag{15}$$

where $\mathrm{qa}(z_i)$ represents the parents of $z_i$ under the proposal, and $\mathbf{z}_{\mathrm{qa}}(i)$ represents all samples of those parents. Importantly, note that the proposals are indexed only by $k_i$, and not by $k_{\mathrm{qa}(z_i)}$, so we can always use the same factorisation structure (Eq. 11) for a factorised and non-factorised proposal. Consult the Appendix for further details, including a proof (Appendix B) and commentary on the choice of approximate posterior (Appendix C).

## 2.4 Computational costs for TMC and IWAE

In principle, TMC could be considerably more expensive than IWAE, as IWAE's cost is linear in $K$, whereas for TMC, the cost scales with $K^T$, where $T$ is the tree-width. Of course, in exchange, we obtain an exponential number of importance samples, $K^n$, so this tradeoff will usually be worthwhile. Remarkably however, the dominant computational costs — those of propagating samples through the neural network — are *linear* in $K$ for typical networks and problem sizes, and hence almost equivalent to that of IWAE. In particular, consider a chained model, where $z = (z_1, z_2, \ldots, z_n)$, and,

$$P(x, z) = P(x|z_1) P(z_1|z_2) \cdots P(z_{n-1}|z_n) P(z_n) \tag{16}$$

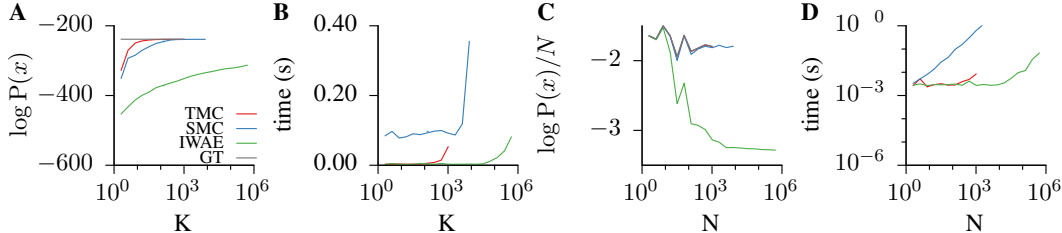

Figure 1: Performance of TMC, SMC, IWAE and ground truth (GT) on a simple Gaussian latent variable example, run in PyTorch using a GPU. **A.** The marginal likelihood estimate (y-axis) for different numbers of particles, $K$ (x-axis), with the number of data points fixed to $N = 128$. **B.** The time required for computing marginal likelihood estimates in **A** on a single Titan X GPU. **C.** The marginal likelihood estimate per data point (y-axis), for models with different numbers of data points, $N$, and a fixed number of particles, $K = 128$. Note that the TMC, SMC and GT lines lie on top of each other. **D.** The time required for computing marginal likelihood estimates in **C**.

In most deep models, the latents, $z_i$, are vectors, and the generative (and recognition) models have the form,

$$P(z_i|z_{i+1}) = \mathcal{N}\left(z_i|\mu_i(z_{i+1}), \text{diag}(\sigma_i^2(z_{i+1}))\right) \tag{17}$$

i.e. the elements of $z_i$ are independent, with means and variances given by neural-networks applied to the activations of the previous layer, $\mu_i(z_{i+1})$ and $\sigma_i^2(z_{i+1})$. As such, the asymptotically quadratic cost of evaluating $P\left(z_i^{k_i}|z_{i+1}^{k_{i+1}}\right)$ for all $k_i$ and $k_{i+1}$ is dominated by the *linear* cost of computing $\mu_i(z_{i+1}^{k_{i+1}})$ and $\sigma_i^2(z_{i+1}^{k_{i+1}})$ for all $k_{i+1}$ by applying neural networks to the activations at the previous layer. Finally, the same considerations apply to non-factorised recognition models (see Appendix): they have the same quadratic asymptotic cost, but linear cost in typical problem sizes.

## 3 Toy Experiments

Here we perform two toy experiments. First, we compare TMC, SMC and IWAE, finding that TMC gives bounds on the log-probability that are considerably better than those for IWAE, and as good (if not better than) SMC, while being considerably faster. Second, we consider an example where non-factorised posteriors might become important. In these toy experiments, we use models in which all variables are jointly Gaussian, which allows us to compute the exact marginal likelihood, and to assess the tightness of the bounds.

### 3.1 Comparing TMC, SMC and IWAE

First, we considered a simple example, with Gaussian parameters, latents and data. There was a single parameter, $\theta$, drawn from a standard normal, which set the mean of $N$ latent variables, $z_i$. The $N$ data points, $x_i$, have unit variance, and mean set by the latent variable,

$$P(\theta) = \mathcal{N}\left(\theta; 0, 1\right), \tag{18}$$
$$P(z_i|\theta) = \mathcal{N}\left(z_i; \theta, 1\right), \tag{19}$$
$$P(x_i|z_i) = \mathcal{N}\left(x_i; z_i, 1\right). \tag{20}$$

For the proposal distributions for all methods, we used the generative marginals,

$$Q(\theta) = \mathcal{N}\left(\theta; 0, 1\right), \tag{21}$$
$$Q(z_i) = \mathcal{N}\left(z_i; 0, 2\right). \tag{22}$$

While this model is simplistic, it is a useful initial test case, because the ground true marginal likelihood can be computed (GT).

We computed marginal likelihood estimates for TMC, SMC and IWAE. First, we plotted the bound on the marginal likelihood against the number of particles for a fixed number of data points, $N = 128$ (Fig. 1A). As expected, IWAE was dramatically worse than other methods: it was still far from the true marginal likelihood estimate, even with one million importance samples. We also found that

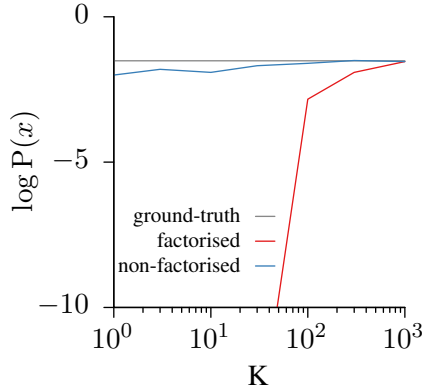

Figure 2: The bound on the log-marginal likelihood for factorised and non-factorised models as compared to the ground-truth, for different number of samples, $K$.

for a fixed number of particles/samples, TMC was somewhat superior to SMC. We suspect that this is because TMC sums over all possible combinations of particles, while the SMC resampling step explicitly eliminates some of these combinations. Further, SMC was considerably slower than TMC as resampling required us to perform an explicit loop over data points, whereas TMC can be computed entirely using tensor sums/products, which can be optimized efficiently on the GPU (Fig. 1B).

Next, we plotted the log-marginal likelihood per data point as we vary the number of data points, with a fixed number of importance samples, $K = 128$ (Fig. 1C). Again, IWAE is dramatically worse than the other methods, whereas both TMC and SMC closely tracked the ground-truth result. However, note that the time taken for SMC (Fig. 1D) is larger than that for the other methods, and scales linearly in the number of data points. In contrast, the time required for TMC remains constant up to around 1000 data points, as GPU parallelisation is exploited increasingly efficiently in larger problems.

### 3.2 Comparing factorised and non-factorised proposals

Non-factorised proposals have a range of potential benefits, and here we consider how they might be more effective than factorised proposals in modelling distributions with very high prior correlations. In particular, we consider a chain of latent variables,

$$P(z_i|z_{i-1}) = \mathcal{N}(z_{i-1}, 1/N), \tag{23}$$
$$P(x|z_N) = \mathcal{N}(z_N, 1), \tag{24}$$

where $z_0 = 0$. As $N$ becomes large, the marginal distribution over $z_N$ and $x$ remains constant, but the correlations between adjacent latents (i.e. $z_{i-1}$ and $z_i$) become stronger. For the factorised proposal, we use the marginal variance (i.e. $Q(z_i) = \mathcal{N}(0, i/N)$), whereas we used the prior for the non-factorised proposal. Taking $N = 100$ (Fig. 2), we find that the non-factorised method considerably outperforms the factorised method for small numbers of samples, $K$, because the non-factorised method is able to model tight prior-induced correlations.

## 4 Experiments

We considered a model for MNIST handwritten digits with five layers of stochastic units inspired by Sønderby et al. (2016). This model had 4 stochastic units in the top layer (furthest from the data), then 8, 16, 32, and 64 units in the last layer (closest to the data). In the generative model, we had two determinstic layers between each pair of stochastic layers, and these deterministic layers had twice the number of units in the corresponding stochastic layer (i.e. 8, 16, 32, 64 and 128). In all experiments, we used the Adam optimizer (Kingma & Ba, 2014) using the PyTorch default hyperparameters, and weight normalization (Salimans & Kingma, 2016) to improve numerical stability. We used leaky-relu nonlinearities everywhere except for the standard-deviations (Sønderby et al., 2016), for which we used $0.01 + \text{softplus}(x)$, to improve numerical stability by ensuring that the standard deviations could not become too small. Note, however, that our goal was to give a fair comparison between IWAE and

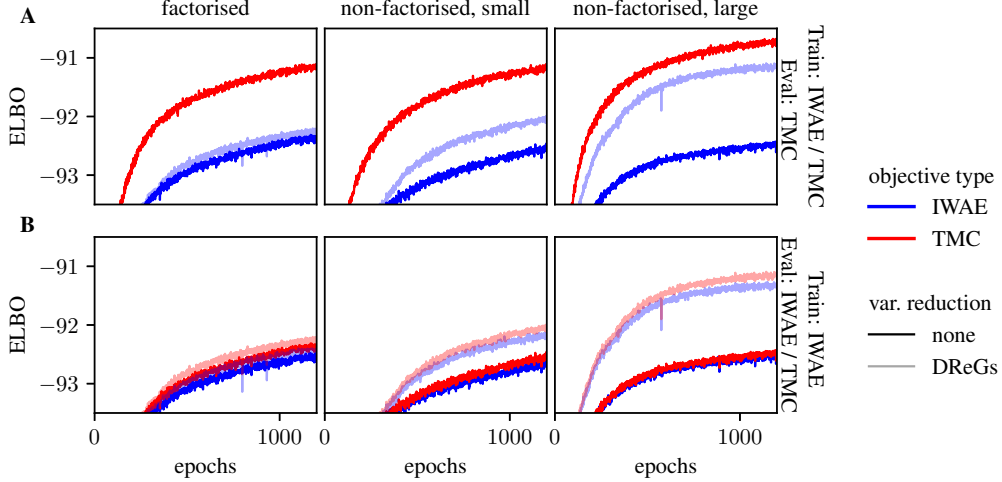

Figure 3: The quality of the variational lower bound for a model of MNIST handwritten digits, with different recognition models and training schemes. We used three different recognition models (columns): factorised (left) where the distribution over the latents at each layer was independent; non-factorised, small (middle) where each stochastic layer depended on the previous stochastic layer through a two-layer deterministic neural network, with a small number of units (the same as in the generative model); and non-factorised, large (right) where the deterministic networks linking stochastic layers in the recognition model had 4 times as many units as in the smaller network. **A.** We trained two sets of models using IWAE (blue) and TMC (red), and plotted the value of the TMC objective for both lines. **B.** Here, we consider only models trained using the IWAE objective, and evalute them under the IWAE objective (blue) and the TMC objective (red).

TMC under various variance reduction schemes, not to reach state-of-the-art performance. As such, there are many steps that could be taken to bring results towards state-of-the-art, including the use of a ladder-VAE architecture, wider deterministic layers, batch-normalization, convolutional structure and using more importance samples to evaluate the model (Sønderby et al., 2016).

We compared IWAE and TMC under three different recognition models, as well as three different variance reduction schemes (including plain reparameterisation gradients). For the non-factorised recognition models (Fig. 3 middle and right), we used,

$$Q(z|x) = Q(z_5|z_4) Q(z_4|z_3) Q(z_3|z_2) Q(z_2|z_1) Q(z_1|x), \tag{25}$$

see Appendix C for the extension to the TMC case. For all of these distributions, we used,

$$Q(z_{i+1}|z_i) = \mathcal{N}\left(\nu_{i+1}, \rho_{i+1}^2\right), \tag{26a}$$

$$\nu_{i+1} = \text{Linear}(h_i), \tag{26b}$$

$$\rho_{i+1} = \text{SoftPlus}\left(\text{Linear}(h_i)\right), \tag{26c}$$

$$h_i = \text{MLP}(z_i). \tag{26d}$$

where the MLP had two dense layers and for the small model (middle), the lowest-level MLP had 128 units, then higher-level MLPs had 64, 32, 16 and 8 units respectively. For the large model, the lowest-level MLP had 512 units, then 256, 128, 64 and 32. For the factorised recognition model,

$$Q(z|x) = Q(z_1|x) Q(z_2|x) Q(z_3|x) Q(z_4|x) Q(z_5|x), \tag{27}$$

we used an architecture mirroring that of the non-factorised recognition model as closely as possible. In particular, to construct these distributions, we used Eqs. (26a–26c), but with a different version of $h_i$ that depended directly on $h_{i-1}$, rather than $z_i$,

$$h_i = \text{MLP}\left(\text{Linear}\left(h_{i+1}\right)\right) \tag{28}$$

where we require the linear transformation to reduce the hidden dimension down to the required input dimension for the MLP.

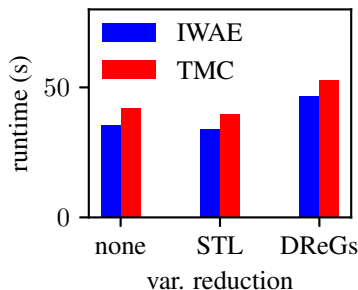

Figure 4: The average time (across the three models) required for one training epoch of the six methods considered above: IWAE/TMC in combination with no additional variance reduction scheme (none), STL, DReGs.

For IWAE and TMC, we considered plain reparametrised gradient descent (none), as well as DReGs (Tucker et al., 2018) (which resolve issues raised by (Rainforth et al., 2018)).

We began by training the above models and variance reduction techniques using an IWAE (blue) and a TMC (red) objective (Fig. 3A). Note that we evaluated both models using the TMC objective to be as generous as possible to IWAE (see Fig. 3B and discussion below). We found that the best performing method for all models was plain TMC (i.e. without DReGs; Fig. 3A). It is unsuprising that TMC is superior to IWAE, because TMC in effect considered $20^5$ importance samples, whereas IWAE considered only 20 importance samples. However, it is unclear whether variance reduction techniques such as DReGs should prove effective in combination with TMC. We can speculate that DReGs performs poorly in combination with TMC because these methods are designed to improve performance as the approximate posterior becomes close to the true posterior (see also Rainforth et al. (2018)). However, TMC considers all combinations of samples, and while some of those combinations might be drawn from the true posterior (e.g. we might have all latents for a particular $k$, $z_1^k, z_2^k, \ldots, z_n^k$, being drawn jointly from the true posterior), it cannot be the case that all combinations are (or can be considered as) drawn jointly from the true posterior. Furthermore, we found that DReGs was numerically unstable in combination with TMC, though it is not clear whether this was an inherent property or merely an implementational issue. Furthermore, note that TMC offers larger benefits over IWAE for the factorised and small non-factorised model, where — presumably — the mismatch between the approximate and true posterior is larger. Next, we considered training the model under just IWAE, and evaluating under IWAE and TMC (Fig. 3B). Evaluating under TMC consistently gave a slightly better bound than evaluating under IWAE, and as expected, because training is based on IWAE, we found that DReGs gave considerably improved performance. As such, in Fig. 3A, we used TMC to evaluate the model trained under an IWAE objective, so as to be as generous as possible to IWAE.

We found that the training time for TMC was similar to that for IWAE, despite TMC considering — in effect — $20^5 = 3,200,000$ importance samples, whereas IWAE considered only 20 (Fig. 4). Further, we found that using STL gave similar runtime, whereas DReGs was more expensive (though it cannot be ruled out that this is due to our implementation). That said, broadly, there is reason to believe that vanilla reparameterised gradients may be more efficient than STL and DReGs, because using vanilla reparameterised gradients allows us to compute the recognition sample and log-probability in one pass. In contrast, for STL and DReGs, we need to separate the computation of the recognition sample and log-probability, so that it is possible to stop gradients in the appropriate places.

## 5 Discussion

We showed that it is possible to extend multi-sample bounds on the marginal likelihood by drawing samples for each latent variable separately, and averaging across all possible combinations of samples from each variable. As such, we were able to achieve lower-variance estimates of the marginal likelihood, and hence better bounds on the log-marginal likelihood than IWAE and SMC. Furthermore,

computation of these bounds parallelises effectively on modern GPU hardware, and is comparable to the computation required for IWAE.

Our approach can be understood as introducing ideas from classical message passing (Pearl, 1982, 1986; Bishop, 2006) into the domains of importance sampling and variational autoencoders. Note that while message passing has been used in the context of variational autoencoders to sum over discrete latents (e.g. Johnson et al., 2016; Bingham et al., 2018), here we have done something fundamentally different: weaving message-passing like approaches into the fabric of importance sampling, by first drawing $K$ samples for each *continuous* latent, and then using message-passing like algorithms to sum over all possible combinations of samples.

### Acknowledgements

I would like to thank HHMI for funding and computational infrastructure.

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
