[Supplementary Material]

## A   Examples of efficient averaging

For instance, consider the generative model in Fig. 5A. The corresponding TMC estimator is

$$\mathcal{P}_{\text{TMC}} = \frac{1}{K_1 K_2 K_3 K_4} \sum_{k_1,k_2,k_3,k_4} f_1^{k_1 k_2} f_2^{k_1 k_3} f_3^{k_2 k_4} f_4^{k_3 k_4}, \tag{29}$$

which can be understood by reference to a loopy factor graph defined over $k_1, k_2, k_3, k_4$ (Fig. 5B). Summing over $k_1$, we obtain Fig. 5C,

$$\mathcal{P}_{\text{TMC}} = \frac{1}{K_2 K_3 K_4} \sum_{k_2,k_3,k_4} f_{12}^{k_2 k_3} f_3^{k_2 k_4} f_4^{k_3 k_4} \tag{30}$$

$$f_{12}^{k_2 k_3} = \frac{1}{K_1} \sum_{k_1} f_1^{k_1 k_2} f_2^{k_1 k_3}, \tag{31}$$

and summing over $k_2$ we obtain Fig. 5D,

$$\mathcal{P}_{\text{TMC}} = \frac{1}{K_3 K_4} \sum_{k_3,k_4} f_{123}^{k_3 k_4} f_4^{k_3 k_4} \tag{32}$$

$$f_{123}^{k_3 k_4} = \frac{1}{K_2} \sum_{k_2} f_{12}^{k_2 k_3} f_3^{k_2 k_4}, \tag{33}$$

which be computed directly. Now we can find the optimal settings for the generative and proposal parameters by performing gradient ascent on $\log \mathcal{P}_{\text{TMC}}$ using standard automatic differentiation tools.

As a second more practical example, consider the generative model in Fig. 6A, with unknown parameters, $\theta$, and unknown latents, $z_i$, corresponding to each data point, $x_i$. The corresponding TMC estimator is,

$$\mathcal{P}_{\text{TMC}} = \frac{1}{K_\theta K^N} \sum_{k_\theta,k_1,k_2,\ldots,k_N} f_\theta^{k_\theta} \prod_{i=1}^{N} f_i^{k_\theta,k_i}, \tag{34}$$

where $k_i$, which runs from 1 to $K$, indexes samples of $z_i$ and $k_\theta$, which runs from 1 to $K_\theta$, indexes samples of $\theta$. We can represent this estimator as a factor graph (Fig. 6B). To efficiently compute the TMC estimate, we sum over $k_1, k_2, \ldots, k_n$,

$$\mathcal{P}_{\text{TMC}} = \frac{1}{K_\theta} \sum_{k_\theta=1}^{K_\theta} f_\theta^{k_\theta} \prod_{i=1}^{N} f_i^{k_\theta} \tag{35}$$

$$f_i^{k_\theta} = \frac{1}{K} \sum_{k_i=1}^{K} f_i^{k_\theta,k_i}, \tag{36}$$

which is represented in Fig. 6C and can be computed directly.

## B   TMC for non-factorised proposals

Now that we have established the possibility of efficiently computing the TMC marginal likelihood estimate, we come back to show that it is possible to use non-factorised proposals in TMC.

Unfortunately, the proof is considerably more involved than the previous proof for factorised TMC, requiring us to consider the joint distribution over all samples for all latents, $\mathbf{z} = (\mathbf{z}_1, \mathbf{z}_2, \ldots, \mathbf{z}_n)$, where all samples for the $i$th latent are given by $\mathbf{z}_i = (z_i^1, z_i^2, \ldots, z_i^{K_i})$. Our approach is to define sets of generative distributions, $P_{\mathbf{k}}(\mathbf{z})$ and $P_{\mathbf{k}}(x|\mathbf{z})$, indexed by $\mathbf{k} = (k_1, k_2, \ldots, k_n)$ such that for all choices of $\mathbf{k}$, the usual importance ratio gives an unbiased estimate of the model evidence,

$$P(x) = E_{Q(\mathbf{z})}\left[ P_{\mathbf{k}}(x|\mathbf{z}) \frac{P_{\mathbf{k}}(\mathbf{z})}{Q(\mathbf{z})} \right] \tag{37}$$

Figure 5: A graphical depiction of the proceedure for efficiently computing the marginal likelihood for a loopy factor graph. **A.** The original graphical model. **B.** Representing the TMC unbiased estimator as a factor graph. **C.** Summing over $k_1$ simplifies the graph. **D.** Summing over $k_2$ gives a simple graph that can readily be summed out.

Figure 6: A graphical depiction of the proceedure for efficiently computing the marginal likelihood for a latent variable model with unknown parameters, $\theta$, and latents, $z_i$ corresponding to each data point, $x_i$. **A.** The original graphical model. **B.** Representing the TMC unbiased estimator as a factor graph. **C.** Summing over $k_1, k_2, \ldots k_n$ simplifies the graph, allowing the TMC estimator to be readily computed by summing over $k_\theta$.

To obtain this equality, we split the full latent space, $\mathbf{z}$ into the "indexed" latents, $z^{\mathbf{k}} = (z_1^{k_1}, z_2^{k_2}, \ldots, z_n^{k_n})$, and the other, "non-indexed" latents, $z^{-\mathbf{k}}$. We chose the likelihood, $P_{\mathbf{k}}(x|\mathbf{z})$, such that the data depends on only the indexed latents, $z^{\mathbf{k}}$, in exactly the same way as in the original model,

$$P_{\mathbf{k}}(x|\mathbf{z}) = P\big(x|z = z^{\mathbf{k}}\big). \tag{38}$$

For the prior, we begin by factorising it into terms for the indexed and non-indexed latents,

$$P_{\mathbf{k}}(\mathbf{z}) = P\big(z^{-\mathbf{k}}|z^{\mathbf{k}}\big) P\big(z^{\mathbf{k}}\big). \tag{39}$$

and we chose the distribution over the indexed latents to be that under the original model,

$$P\big(z^{\mathbf{k}}\big) = P\big(z = z^{\mathbf{k}}\big). \tag{40}$$

These two choices are all that is required to give an unbiased estimator of the original model evidence. In particular,

$$E_{Q(\mathbf{z})}\left[P_{\mathbf{k}}(x|\mathbf{z}) \frac{P_{\mathbf{k}}(\mathbf{z})}{Q(\mathbf{z})}\right] = \int dz^{\mathbf{k}}\, dz^{-\mathbf{k}}\, P\big(x|z^{\mathbf{k}}\big) P\big(z^{\mathbf{k}}\big) P\big(z^{-\mathbf{k}}|z^{\mathbf{k}}\big) \tag{41}$$

integrating over $z^{-\mathbf{k}}$, then using our choices for the likelihood and prior,

$$E_{Q(\mathbf{z})}\left[P_{\mathbf{k}}(x|\mathbf{z}) \frac{P_{\mathbf{k}}(\mathbf{z})}{Q(\mathbf{z})}\right] = \int dz^{\mathbf{k}}\, P\big(x|z^{\mathbf{k}}\big) P\big(z^{\mathbf{k}}\big) = \int dz\, P(x|z) P(z) = P(x), \tag{42}$$

as required. Importantly, note that this derivation made no assumptions about the proposal, $Q(\mathbf{z})$, and the generative model for the non-indexed latents, $P\big(z^{-\mathbf{k}}|z^{\mathbf{k}}\big)$, giving us complete freedom — at least in principle — about how we choose those quantities.

However, importance sampling over the enlarged latent space ($\mathbf{z}$) may give rise to higher variance estimators than working in the orignal space, ($z$ or $z^{\mathbf{k}}$). As such, it pays to be careful about the choice of generative model for the non-indexed latents, $\mathrm{P}\left(z^{-\mathbf{k}}|z^{\mathbf{k}}\right)$, and the proposal, $\mathrm{Q}(\mathbf{z})$. In particular, our strategy is to choose $\mathrm{P}\left(z^{-\mathbf{k}}|z^{\mathbf{k}}\right)$ such that it cancels many of the terms in $\mathrm{Q}(\mathbf{z})$. We begin by assuming that each sample for a single latent is independent, conditioned on all samples of previous latents,

$$\mathrm{Q}(\mathbf{z}|x) = \prod_i \prod_{k_i} \mathrm{Q}\left(z_i^{k_i}|x, \mathbf{z}_{\mathrm{qa}(i)}\right) \tag{43}$$

where $\mathrm{qa}(z_i) \subseteq \{1, \ldots, n\}$ gives the indices of the parents of $z_i$ under the proposal. To give as much cancellation as possible, we assume that the generative model for the non-indexed latents is equal to the proposal,

$$\mathrm{P}\left(z^{-\mathbf{k}}|z^{\mathbf{k}}\right) = \prod_i \prod_{k_i' \neq k_i} \mathrm{Q}\left(z_i^{k_i'}|x, \mathbf{z}_{\mathrm{qa}(i)}\right) \tag{44}$$

After cancelling $\mathrm{P}\left(z^{-\mathbf{k}}|z^{\mathbf{k}}\right)$ with terms in the proposal the importance ratio becomes,

$$\mathrm{P}_{\mathbf{k}}(x|\mathbf{z}) \frac{\mathrm{P}_{\mathbf{k}}(\mathbf{z})}{\mathrm{Q}(\mathbf{z}|x)} = \frac{\mathrm{P}\left(x|z^{\mathbf{k}}\right)\mathrm{P}\left(z^{\mathbf{k}}\right)}{\prod_i \mathrm{Q}\left(z_i^{k_i}|x, \mathbf{z}_{\mathrm{qa}(i)}\right)} = \frac{\mathrm{P}\left(x, z_1^{k_1}, z_2^{k_2}, \ldots, z_n^{k_n}\right)}{\prod_i \mathrm{Q}\left(z_i^{k_i}|x, \mathbf{z}_{\mathrm{qa}(i)}\right)} \tag{45}$$

Notably, this is analogous to the factorised case, except that the proposal is allowed to depend on samples of the other latents. Further, note that the techniques for efficient averaging continue to work in exactly the same way: the proposal factors depend only on one index, $k_i$, and so we can always use the same factorisation of the importance ratio under a factorised or non-factorised approximate posterior.

## C   Typical choice of approximate posteriors

In the previous section, we used approximate posteriors of the form, $\mathrm{Q}\left(z_i^{k_i}|x, \mathbf{z}_{\mathrm{qa}(i)}\right)$. In general it is possible to use any (permutation invariant) function of the set of previous samples, for instance, taking their mean, and this would be computationally efficient. However, in order to match TMC and IWAE as closely as possible, it is appropriate to use an mixture distribution, where each component of the mixture model depends on one combination of past samples,

$$\mathrm{Q}\left(z_i^{k_i}|x, \mathbf{z}_{\mathrm{qa}(i)}\right) = \sum_{k_{\mathrm{qa}(i)}} \mathrm{Q}\left(z_i^{k_i}|x, z_{\mathrm{qa}(i)}^{k_{\mathrm{qa}(i)}}\right). \tag{46}$$

In our experiments, the non-factorised distributions depend only on one previous random variable, so they become,

$$\mathrm{Q}\left(z_i^{k_i}|x, \mathbf{z}_{\mathrm{qa}(i)}\right) = \sum_{k_{i-1}} \mathrm{Q}\left(z_{i-1}^{k_{i-1}}|z_{i-1}^{k_{i-1}}\right). \tag{47}$$

Again, the asymptotic cost of sampling and evaluating the log-probability for all $k_i$ is $\mathcal{O}(K^2)$, but for practical problem sizes, the dominant cost is computing the mixture components, $\mathrm{Q}\left(z_{i-1}^{k_{i-1}}|z_{i-1}^{k_{i-1}}\right)$ by propagating the previous samples, $z_{i-1}^{k-1}$, through the appropriate neural network, and this cost again scales with $\mathcal{O}(K)$.

## D   Exact marginalisation over discrete latent variables

Notably, the TMC framework can be extended to incorporate exact marginalisation over discrete variables. In particular, we take the number of importance samples, $K_i$, to be equal to the number of settings of the discrete variable, we consider a uniform proposal, $\mathrm{Q}(z_i) = 1/K_i$, and we use stratified sampling, such that each possible setting of the latent variable is represented by one sample (e.g.

taking $z_i = \{1, 2, \ldots, K_i\}$, we might have $z_i^{k_i} = k_i$). Making these choices, and taking $z_1$ to be a discrete variable, the TMC estimator over $\mathrm{P}\left(x, z_1^{k_1}, z_2^{k_2}, \ldots, z_n^{k_n}\right)$ is,

$$\mathcal{P}_{\mathrm{TMC}} = \frac{1}{K^{n-1}} \sum_{k_2, \ldots, k_n} \sum_{z_1} \frac{\mathrm{P}\left(x, z_1, z_2^{k_2}, z_3^{k_3} \ldots, z_n^{k_n}\right)}{\mathrm{Q}\left(z_2^{k_2}\right) \mathrm{Q}\left(z_3^{k_3}\right) \cdots \mathrm{Q}\left(z_n^{k_n}\right)}. \tag{48}$$

Note that this is exactly equal to a different TMC estimator, over a model with $z_1$ marginalised out (i.e. $\mathrm{P}(x, z_2, z_3, \ldots, z_n)$). This is important because it enables us to link TMC with the rich prior literature on exact marginalisation in discrete graphical models, and because it allows us to combine importance sampling over continuous variables and exact marginalisation over discrete variables into a single framework.

## E    Numerically stable matrix (tensor) products in the log-domain

When we take inner products of tensors representing large probabilities, there is a considerable risk of numerical overflow. To avoid this risk, we work in the log-domain, and write down a numerically stable matrix-inner product, denoted logmmexp, by analogy with the standard logsumexp function. In particular, consider the problem of computing $e^{Z_{ik}}$, as the matrix product of $e^{X_{ij}}$ and $e^{Y_{jk}}$,

$$e^{Z_{ik}} = \sum_j e^{X_{ij}} e^{Y_{jk}}. \tag{49}$$

taking the logarithm so as to compute $Z_{ik}$,

$$Z_{ik} = \log\left(\sum_j e^{X_{ij}} e^{Y_{jk}}\right). \tag{50}$$

as the elements of $X_{ij}$ and $Y_{jk}$ could be very large (or very small), to ensure numerical stability of the sum, we add and subtract $x_i$ and $y_k$,

$$Z_{ik} = \log\left(\sum_j e^{X_{ij}-x_i} e^{Y_{jk}-y_k}\right) + x_i + y_k, \tag{51}$$

where

$$x_i = \max_j X_{ij} \tag{52}$$

$$y_k = \max_j Y_{jk}. \tag{53}$$

## F    Combining DReGs and TMC

To perform DReGs, we optimize the generative parameters using the usual IWAE/TMC cost function, but use a different strategy for optimizing the recognition model, that involves non-trivial manipulations of the importance weights. In particular, the DReGs recognition updates are given by,

$$\sum_i \frac{w_i^2}{(\sum_j w_j)^2} \frac{\partial z_i}{\partial \phi} \frac{\partial \log w_i}{\partial z_i} \tag{54}$$

$$= \sum_i \frac{w_i^2}{(\sum_j w_j)^2} \frac{\partial z(\epsilon_i; \phi)}{\partial \phi} \left.\frac{\partial \log w(z; \phi)}{\partial z}\right|_{z=z(\epsilon_i;\phi)},$$

where the first version is that given in prior work, and the second version has been written out more carefully to highlight the functional dependencies, and hence how the partial derivative applies to each term. The latents have been written in their usual reparameterised form, and the importance weights can be written as a function of the latents, and the parameters,

$$w(z; \phi) = \frac{\mathrm{P}(x, z)}{\mathrm{Q}_\phi(x, z)}, \tag{55}$$

but we write down the individual importance weights as functions of the reparameterised noise, $\epsilon_i$, and the parameters,

$$w_i(\epsilon_i; \phi) = w(z(\epsilon_i; \phi); \phi). \tag{56}$$

We cannot compute this function directly in the TMC set-up, because TMC involves exponential numbers of importance samples, and allows only a fairly restricted set of operations (such as summing the importance weights) to be performed efficiently. In contrast, DReGs appears to require complex, almost arbitrary operations over the importance weights. However, it is possible to write down a surrogate objective, utilizing the stop-gradients operation, that does have the required gradients. To do so, we need to begin by carefully introducing notation. Remember that any particular importance weight, $w_i$, can be written as a function of the uniform noise, $\epsilon_i$ and the parameters (Eq. (56)), and as such, the gradient of $w_i$ can be broken up into two terms,

$$\frac{\partial w_i(\epsilon_i; \phi)}{\partial \phi} = \frac{\partial w(z_i; \phi)}{\partial \phi} + \frac{\partial z(\epsilon_i; \phi)}{\partial \phi} \left. \frac{\partial w(z; \phi)}{\partial z} \right|_{z=z(\epsilon_i;\phi)} \tag{57}$$

where the first term is the direct effect of $\phi$, on $w_i$, and the second term is the "indirect" effect, through the reparameterised latents. Now we are in a position to define $\hat{w}_i$ and $\bar{w}_i$, which always have the same value as $w_i$, but where we have applied the stop-gradients operation to drop different gradient terms. In particular, $\bar{w}_i$ stops all gradients, so that (in a slight abuse of notation),

$$\frac{\partial \bar{w}_i}{\partial \phi} = 0, \tag{58}$$

and $\hat{w}_i$ stops only the "direct" term in Eq. (57), such that,

$$\frac{\partial \hat{w}_i}{\partial \phi} = \frac{\partial z(\epsilon_i; \phi)}{\partial \phi} \left. \frac{\partial w(z; \phi)}{\partial z} \right|_{z=z(\epsilon_i;\phi)} \tag{59}$$

Now, we hypothesise that the DReGs estimator can be rewritten as,

$$\frac{1}{2} \left( \frac{\sum_j \bar{w}_j^2}{(\sum \bar{w}_j)^2} \right) \log \sum_i \hat{w}_i^2 \tag{60}$$

Note that this is written entirely in terms of $\sum_j w_j$, which can be computed directly using TMC, as given above, and $\sum_j w_j^2$, which can be computed by running TMC again, with squared weights. The gradient of $\bar{w}_j$ is zero, so

$$\frac{\partial}{\partial \phi} \left[ \frac{1}{2} \left( \frac{\sum_j \bar{w}_j^2}{(\sum \bar{w}_j)^2} \right) \log \sum_i \hat{w}_i^2 \right] \tag{61}$$

$$= \frac{1}{2} \left( \frac{\sum_j w_j^2}{(\sum w_j)^2} \right) \frac{\partial}{\partial \phi} \log \sum_i \hat{w}_i^2, \tag{62}$$

where the value of $w_j$, $\hat{w}_j$ and $\bar{w}_j$ is equal, so when the gradient operation is no longer applied, we can revert to the standard notation, $w_j$. Applying the derivative to the logarithm,

$$= \frac{1}{2} \left( \frac{\sum_j w_j^2}{(\sum w_j)^2} \right) \frac{\sum_i \frac{\partial}{\partial \phi} \hat{w}_i^2}{\sum_j w_j^2}, \tag{63}$$

cancelling terms,

$$= \frac{1}{2} \frac{\sum_i \frac{\partial}{\partial \phi} \hat{w}_i^2}{(\sum w_j)^2}, \tag{64}$$

and applying the derivative to $\hat{w}_j^2$,

$$= \frac{\sum_i w_i \frac{\partial}{\partial \phi} \hat{w}_i}{(\sum w_j)^2}, \tag{65}$$

Figure 7: Performance of TMC, SMC, IWAE and ground truth (GT) on a simple Gaussian latent variable example, run in PyTorch using a GPU. **A.** The marginal likelihood estimate (y-axis) for different numbers of particles, $K$ (x-axis), with the number of data points fixed to $N = 128$. **B.** The time required for computing marginal likelihood estimates in **A** on a single Intel Xeon compute core. **C.** The marginal likelihood estimate per data point (y-axis), for models with different numbers of data points, $N$, and a fixed number of particles, $K = 128$. Note that the TMC, SMC and GT lines lie on top of each other. **D.** The time required for computing marginal likelihood estimates in **C**.

Finally, using $\frac{\partial}{\partial\phi}\hat{w}_i = w_i \frac{\partial}{\partial\phi}\log\hat{w}_i$,

$$= \sum_i \frac{w_i^2}{\left(\sum_j w_j\right)^2}\frac{\partial}{\partial\phi}\log\hat{w}_i, \tag{66}$$

and substituting the gradient of $\hat{w}_i$,

$$= \sum_i \frac{w_i^2}{\left(\sum_j w_j\right)^2}\frac{\partial z_i}{\partial\phi}\frac{\partial\log w_i}{\partial z_i}, \tag{67}$$

which matches Eq. (54), as required.

## G    TMC helps even on CPU

To confirm that our results were not just applicable to GPUs, we redid Fig. 1 using a single-threaded CPU implementation. Replicating Fig 1BD, we find similar, albeit less extreme results, with TMC always being faster than SMC. This conflicts with the asymptotic results, which suggest that TMC ($O(K^2)$) should be slower than SMC ($O(K)$). This conflict likely arises because of the overhead of the Python interpreter: in this model with TMC, we can perform the reduction over all $N$ latent variables in a single efficient tensor operation (on one CPU), whereas the SMC resampling step requires us to use an explicit Python for-loop.

## H    Complex, random graphical model

The latents, $z_i$ were drawn IID from a standard Gaussian,

$$P(z_j) = \mathcal{N}\left(z_j; 0, 1\right) \tag{68}$$

and the data, $x_i$ was generated by,

$$P(x_i|z_1,\dots,z_N) = \mathcal{N}\left(x_i; \sum_j C_{ij}z_j, 0.1^2\right) \tag{69}$$

where C is a fixed, known binary matrix,

$$P(C_{ij}) = \text{Bernoulli}(0.01). \tag{70}$$

While the prior in the model is simple, the posterior factor graph has complex, random structure due to explaining-away: in our experiment, the factor graph had a tree width of 4. Even 16 million IWAE importance samples, (15 s on CPU) is insufficient (marginal likelihood estimate is around $-1400$). In contrast, only $K = 64$ TMC importance samples were required (0.4 s on CPU), giving a marginal likelihood estimate of $-69.9$, compared to a true value of $-67.1$. Ultimately, these asymptotic results imply that, if the number of latents is considerably greater than the tree width (i.e. there are exploitable conditional independencies), then TMC will give dramatic benefits over IWAE.