[Reviews · NeurIPS 2019]

Reviewer 1



Summary: The authors describe an improved objective function for variational inference. In the spirit of the Importance Weighted Autoencoder they use multiple samples from the approximating distribution to obtain a tighter bound on the log marginal probability. The key insight of the paper is that they can use all combinations (across subsets of parameters) of samples drawn from the approximating distribution to compute marginal probability estimator. Naively this would require computation that scales exponentially in the number of subsets. The authors reduce this complexity by exploiting dependency structures in the generative model. They show the method is applicable when using fully factorized approximating distributions, or more complex approximating distributions which preserve some dependency structure. The authors benchmark their approach on synthetic data sets and compare the accuracy of estimates for the log marginal to IWAE and variational SMC. They finish by exploring the use of two variance reduction schemes with their method and IWAE. Comments: - The paper is reasonably well written. The introduction and background are relevant. The description of the approach is reasonably easy to follow, though some important points are in the supplemental. In particular, the complexity non-factorized method is not really presented in the main text. The non-factorized version in fact uses a different generative model in the objective, which is not obvious from the main text. - The idea is quite interesting and is a nice synthesis of ideas from the VAE, graphical models and MCMC literature. I feel like there may also be some interesting connections to the discrete particle filter literature. - The paper focuses on the methods ability to provide a tighter bound on the marginal likelihood. It is not clear that this leads to any performance improvement in the training of the neural nets. - The discussion of computational complexity is hard to follow. If understand correctly the method will scale exponentially in the size of the largest factor. The authors argue this is unimportant in the context of neural nets. They only show a simple linear chain model, so it has no impact on performance. But in other models it seems likely that it will make their approach slower than the IWAE, possibly significantly. - The authors seem to make a major point of how the method meshes well with tensor implementation on GPUs. To me that is a minor point, as the major insight is the reduction in computational complexity that is achieved by using the sum-product algorithm. - Figure 3 is hard/impossible to read. Better color coding or line style choice is needed. - (Post author feedback) The authors have proposed to address most of of my concerns in the final manuscript. Thus I support publication.

Reviewer 2



First of all, this paper is hard to understand. Many important discussion and results are shown only in the supplemental material and the author does not specify in the main paper that which section in the Appendix we should refer. I cannot follow how Eq (11) and Eq (12) are derived under what kind of models. The author just mentioned that Eq (11) holds for "In this (and many other cases)", but I cannot understand what this sentence means. Also, there is no formal definition of kapper_j in the main paper (I found it in Eq (36) in the appendix). The experiments are not enough. Although in the second paragraph of the Introduction, the author indicated that IWAE is not enough for large models, the experiments are only conducted on very simple models which IWAE works well. I think additional experiments in real-world data in large models are needed to verify the usefulness of the proposed method.

Reviewer 3



EDIT POST-REBUTTAL I maintain my original score for the reasons already indicated. I think there is value in publishing this work. *** This work is in a long line of papers seeking to improve stochastic optimization for variational inference with a reparameterized lower bound. While the particular algorithmic contribution is not a major departure from the state of the art, the careful empirical study, theoretical ground, and discussion in relation to recent works has made me reconsider what I considered obvious before reading--that DReGS was the most efficient method of gradient estimation due to unbiasedness and lower variance. I appreciate this insight. The writing is very clear, and addresses current open issues in approximate inference. The empirical results are appropriate, using both synthetic and MNIST data.

[Author Response · NeurIPS 2019]

We would like to thank the reviewers for their kind and thoughtful comments.

**Reviewer #1** We have simplified Figure 3 considerably, removing STL (which uses biased gradients), and removing row C.

1) We have updated the paper to clarify that the time and space complexity is $O(K^T)$ for TMC, and $O(K)$ for IWAE and TMC, where $K$ is the number of importance samples per latent, and $T$ is the treewidth (closely related to the number of latent variables associated with the largest factor). So in terms of asymptotic time and space for a single marginal-likelihood estimate, TMC is the most expensive. However, TMC has considerable advantages over both IWAE and SMC. In particular, while IWAE gives only $K$ importance samples, TMC gives $K^n$, where $n$ is the number of latent variables. As such, the key metric is the number of importance samples per unit time (IWAE: $O(1)$, TMC: $O(K^{n-T})$). In the worst case $T = n$ so TMC is the same as IWAE, but more usually $T \ll n$ so TMC dramatically improves over IWAE. The plain bootstrap SMC algorithm is also $O(K)$, and does far better than IWAE, but has two problems that do not occur in TMC. First, the resampling step is non-differentiable (giving biased or high-variance gradients) and forces sequential structure on the computation. Second, the bootstrap particle filter displays particle degeneracy, so we will usually end up with only one sample for the earlier latent variables. Any attempt to mitigate particle degeneracy (e.g. using backward sampling) has the same complexity as TMC, but still has the problems associated with resampling.

2) We have included an example with 50 univariate Gaussian latent variables, and with a randomly connected undirected graph with a tree-width of 4 (details in revised paper). Even 16 million IWAE importance samples, (15 s on CPU) is insufficient (marginal likelihood estimate is $\sim -1400$). In contrast, only $K = 64$ TMC importance samples were required (0.4 s on CPU), giving a marginal likelihood estimate of $-69.9$, compared to a true value of $-67.1$. Ultimately, the asymptotic results described above imply that, if the number of latents is considerably greater than the tree width (i.e. there are exploitable conditional independencies), then TMC will give dramatic benefits over IWAE.

3) We have redone Fig. 1 (which is not a linear chain, see Appendix Fig. 6A) using a single-threaded CPU implementation. Replicating Fig 1BD, we find similar, albeit less extreme results, with TMC always being faster than SMC. This conflicts with the asymptotic results, which suggest that TMC ($O(K^2)$) should be slower than SMC ($O(K)$). This conflict likely arises because of the overhead of the Python interpreter: in this model with TMC, we can perform the reduction over all $N$ latent variables in a single efficient tensor operation (on one CPU), whereas the SMC resampling step requires us to use an explicit Python for-loop.

4) SMC gives almost exactly the same predictive performance as TMC in SSM/HMM's but is slightly faster ($O(K)$ rather than $O(K^2)$). This is because SMC is very well-suited to SSM/HMMs: the advantages of TMC arise in more complex models and in training recognition models (described above and in the paper).

**Reviewer #2** Thanks for the comments on clarity: following your suggestions we have substantially updated the manuscript. In particular, we have included Eq. 36 in the main text, and also included the corresponding choice of factors. This should help to clarify that Eq. 11 applies to any directed graphical model (we have also included references to the well-understood relationships between directed graphical models and factor graphs, e.g. Bishop 2006). We have also stated explicitly that Eq. 12 arises by substituting Eq. 11 into Eq. 9. Finally, thanks for pointing out that we had not referenced sections in the Appendix: this is now fixed.

In the example in Fig. 1, we consider a model that does not have a chain-structure (see Appendix Fig. 6A). In this case, IWAE performs arbitrarily badly due to the high-dimensionality of the state-space. Further, see the Reviewer #1 (2) for details of an additional experiment with a complex, random graphical model structure.

**Reviewer #3** We agree, the connections to DReGS are extremely interesting. We had also expected that DReGS + TMC (see Appendix G for derivation) would give the best performance (and indeed, it would only strengthen the paper if that were the case). However, in our experiments, we found DReGs + TMC to be numerically unstable, despite careful effort in using numerically stable operations (e.g. see Appendix F), and sharing code between vanilla TMC and DReGs + TMC. This numerical instability is particularly problematic because standard tricks to cope with instability (e.g. gradient clipping) give biased gradients and appeared to degrade performance. We are currently investigating these issues. In particular, the DReGs proofs and results seem to suggest that as we use a very large number of importance samples ($K^n$), we should have high-SNR gradients, and hence have numerically stable, rapid learning. However, in TMC, the importance samples (in the full joint space) are not independent — they are highly structured — with only $K$ different values for each latent variable, and we are currently investigating the hypothesis that this structure interacts poorly with DReGs.

We have put the anonymised code here. This is special-case code build around each experiment: my intention is to build an open-source probabilistic programming language around TMC.

[Meta-Review · NeurIPS 2019]

The majority of reviewers are in support of publishing this work, in spite of some of the limitations that they raised and discussed based on the authors' feedback. I recommend that the authors take these comments to heart in updating the manuscript for the camera-ready.